# Lifetime Cadmium Exposure and Mortality for Renal Diseases in Residents of the Cadmium-Polluted Kakehashi River Basin in Japan

**DOI:** 10.3390/toxics8040081

**Published:** 2020-10-01

**Authors:** Muneko Nishijo, Kazuhiro Nogawa, Yasushi Suwazono, Teruhiko Kido, Masaru Sakurai, Hideaki Nakagawa

**Affiliations:** 1Department of Public Health, Kanazawa Medical University, Uchinada, Ishikawa 920-0293, Japan; 2Department of Occupational and Environmental Medicine, Graduate School of Medicine, Chiba University, Chuoku, Chiba 260-8670, Japan; nogawa@chiba-u.jp (K.N.); suwa@faculty.chiba-u.jp (Y.S.); 3Department of Community Health Nursing, School of Health Sciences, Kanazawa University, Kanazawa, Ishikawa 920-0942, Japan; tkido@staff.kanazawa-u.ac.jp; 4Department of Social and Environmental Medicine, Kanazawa Medical University, Uchinada, Ishikawa 920-0293, Japan; m-sakura@kanazawa-med.ac.jp (M.S.); hnakagaw@kanazawa-med.ac.jp (H.N.)

**Keywords:** cadmium, mortality, lifetime cadmium intake, renal diseases, urinary cadmium, a follow-up study

## Abstract

Very few studies have investigated the dose–response relationship between external cadmium (Cd) exposure and mortality. We aim to investigate the relationship between lifetime Cd intake (LCd) and mortality in the Cd-polluted Kakehashi River basin in Japan. Mortality risk ratios for a unit of increase of LCd and urinary Cd were analyzed using Cox’s proportional model. LCd was estimated based on residency and Cd in rice produced in their living areas. In men, mortality for all causes was significantly increased for a 10-μg/g Cr increase in urinary Cd, but not for a 1-g increase in LCd. In women, mortality risks for all causes and renal diseases, particularly renal failure, were significantly increased for a 10-μg/g Cr increase in urinary Cd. Similarly, mortality risks for renal diseases and renal failure were significantly increased for a 1-g increase of LCd in women. Comparing the contribution of two exposure markers to increased mortality in women, LCd was more effective for increasing mortality risks for renal diseases and renal failure, while urinary Cd contributed more to increased mortality risk for all causes. LCd may show a better dose–response relationship with mortality risk for renal diseases in women.

## 1. Introduction

The kidney is a target organ of external cadmium (Cd), and renal tubular dysfunction is the most prevalent adverse health effect induced by Cd exposure. Significant associations between renal tubular dysfunction, indicated by increasing urinary low molecular protein and Cd exposure markers such as urinary Cd, were reported in residents of Cd-polluted areas in Japan [1].

Cd in rice (RCd) is believed to be a good exposure marker in Japan because Tsuchiya and Iwao (1978) [2] reported that Japanese residents took half or two-thirds of Cd from rice in a study conducted in the 1970s. Nogawa et al. (1989) [3] showed a prevalence of renal tubular dysfunction indicated by an increase in urinary β2–microglobulin (β2–MG) ≥1000 µg/g creatinine (g Cr) with increasing residential duration (years) as well as RCd produced in the communities, suggesting that lifetime Cd intake (LCd), defined from both factors of RCd and residential duration, is a useful indicator of external dose to show the dose–response relationship, with health effects induced by Cd exposure. Significant dose–response relationships between LCd and urinary metallothionein in the Cd-polluted Kakehashi River basin [4,5] and between LCd and urinary glucoproteinuria in the Jinzu River basin [6] were reported in residents that were environmentally exposed to Cd in Japan. 

In our previous studies, following-up subjects from the Kakehashi River basin who participated in health impact surveys in 1981–1982 for 9 and 15 years, we reported increased mortality for all causes, cardiovascular diseases, and renal diseases in the subjects with Cd-induced renal tubular dysfunction indicated by urinary β2-MG, protein, glucose, amino acids, and retinol-binding protein [7,8,9,10,11,12]. An investigation of the relationship between the urinary Cd levels and mortality in the 15- and 22-year follow-up studies showed increased mortality for all causes, including renal diseases and heart failure, among subjects with urinary Cd levels ≥10 μg/g Cr in the 1981–1982 survey compared with subjects with Cd levels <3 μg/g Cr [13,14]. 

Relationships between LCd and mortality have been reported in the inhabitants of the Cd-polluted Jinzu River basin in three studies [15,16,17], but these relationships have not been demonstrated among residents of the Kakehashi River basin in our previous studies. Therefore, we extended the follow-up period to 35 years and analyzed dose–response relationships between LCd and mortality for causes of deaths to clarify the effect of environmental Cd exposure on mortality risks, particularly for renal diseases.

## 2. Materials and Methods 

The Kakehashi River basin is one of the Cd-polluted areas in Japan, and it includes 700 ha of rice paddy fields. The pollution was due to mining activity that included a full-scale operation near the Kakehashi River that started in 1930 and continued until mining ceased in 1971. Rice with Cd > 0.4 (ppm) was detected in 12 of 22 villages in this area, and restoration of polluted paddy fields was undertaken from 1977 to 1988.

A total of 2602 subjects ≥50 years of age (1169 men and 1433 women) living in the Cd-polluted Kakehashi River basin, whose residential periods were ascertained in the 1981–1982 health impact survey, were enrolled in the present follow-up study (82% participant rate). LCd was estimated for them from RCd levels produced in their living communities and residential period (days) until 1981–1982 using the following formula: LCd = (RCd × 333.5 g + 34 µg) × residential period in Kakehashi River basin + 50 µg × residential period in nonexposed areas [3] on the assumption of 333.5 g for daily rice intake obtained by duplicated diet method, 34 µg for daily Cd intake from food other than rice from the polluted area [18], and 50 µg for daily Cd intake from nonpolluted areas [19]. 

Their mean levels of age, LCd, urinary Cd, and urinary β2–MG, examined at the survey in 1981–1982, with test results by *t*-test to compare them between sexes, are shown in Table 1. At this time, urinary Cd and β2-MG were transformed to log10 values for analysis because of the improvement of distribution. Positive rates for urinary Cd and urinary β2–MG are also shown in Table 1. Significantly greater mean values of LCd, urinary Cd, and urinary β2–MG and greater positive rates of urinary Cd and β2–MG were found in women compared with men. 

Cd in urine samples was measured by flameless atomic absorption spectrometry after ashing with HNO3, HsSO4, and HClO4 and extraction with ammonium pyrrolidine dithiocarbamate (APDC) and methyl isobutyl ketone (MIBK) [20]. Freeze-dried standard reference material for toxic elements in urine (The National Bureau of Standards, Washington, DC, USA) was used to test the accuracy and precision of the analytical method of urinary Cd. Cd concentration corrected by Cr in urine was used for analysis.

A 35-year prospective follow-up survey of subjects was conducted from the day of their initial examination at the health impact survey in 1981–1982 until November 2016. The survival status (alive or dead), the date of death, place of residence (still residing or not in the target area), and the date of death, if applicable, were determined for 2527 residents (1149 men and 1378 women) from family registry records of all subjects with the cooperation of the Prefecture Public Health Office and City Municipal Office. Then, individual causes of death were ascertained from vital statistics by linking them to health survey data, with survival status based on birthday, death day, gender, and address after getting the permission of the Ministry of Health in Japan. The final number of subjects available for mortality analysis was 2496 (1135 men and 1361 women, 96% follow-up rate) and their mean follow-up period (months) are shown in Table 1. Causes of deaths were classified according to the 9th and 10th Revised International Classification of Diseases. The study protocol was approved by the Ethics Committee of Kanazawa Medical University (No-212, 17 June 2014).

Increased mortality risks for all causes and major causes of deaths, including renal diseases and renal failure, associated with a 1-g increase of LCd or a 10-µg/gCr (1 of log10 transformed value) increase of urinary Cd were investigated after adjusting for age using Cox’s proportional hazard model. This model is one of regression analysis used for investigating associations between the survival time from the date of the baseline survey to the endpoint of subjects and relevant factors. In addition, to investigate which exposure marker, LCd or urinary Cd, contributed more to increased mortality for all causes and renal diseases, the stepwise elimination method based on Wald was used in Cox’s mortality risk analysis. The SPSS (version 21.0) software package for Windows (SPSS Inc., Armonk, NY, USA) was used for statistical analysis.

## 3. Results

### 3.1. Mortality Risk Ratios and Cd Exposure Markers in Men

Hazard ratios for all causes, major causes of deaths, and renal diseases for a 10-μg/g Cr increase of urinary Cd and those for a 1-g increase of LCd after controlling for age in men are shown in Table 2. For men, the hazard ratio for all causes of deaths was significantly increased for a 10-μg/g Cr increase of urinary Cd, albeit no major causes of deaths were found to contribute to increased mortality. For a 1-g increase of LCd, no significantly increased mortality risks for all causes and no causes of death were noted in men (Table 2).

### 3.2. Mortality Risk Ratios and Cd Exposure Markers in Women

In women, hazard ratio was significantly increased for all causes for 10 μg/g Cr increase of urinary Cd. For renal diseases, particularly for renal failure, hazard ratios were significantly increased for10 μg/g Cr increase of urinary Cd in women (Table 3). Although hazard ratio for all causes was not significantly increased for 1 g increase of LCd, mortality risks for renal diseases and renal failure were significantly increased for 1 g increase of LCd in women (Table 3). These results in women indicate that increasing dose of Cd exposure may increase mortality for renal diseases, particularly renal failure. At the same time, however, they rise a question which exposure marker, urinary Cd or LCd, is a better marker to contribute to increasing risk for renal disease deaths. 

### 3.3. Comparisons of Mortality From Renal Diseases between Men and Women at Different Cd Exposure Levels 

Crude mortality rates for renal diseases in six groups with a 1-g difference of LCd (a) and in three groups with a 10-μg difference of urinary Cd (b) in men and women are illustrated in Figure 1.

Crude mortality from renal diseases was higher in men in the lowest exposure group, indicated by LCd (Figure 1a) or urinary Cd (Figure 1b), suggesting that factors other than Cd exposure, such as smoking and hypertension, relevant to chronic kidney disease (CKD), might influence mortality. While we found more deaths for renal diseases in the highest LCd group (LCd ≥ 5 g) in women compared men (Figure 1a), no stable result was obtained in the highest urinary Cd group (urinary Cd ≥ 10 μg/gCr), because of the small number of male subjects. These results suggest that mortality risk ratios for renal diseases might be changed, particularly in men, if relevant factors other than age could be controlled for in the analysis using Cox’s proportional model. 

### 3.4. Comparisons of Effects on Mortality between Urinary Cd and LCd in Women

To investigate which Cd exposure marker contributed to increased mortality from all causes and renal diseases, including renal failure, the stepwise elimination method was used for mortality risk analysis with three explanatory valuables, namely, urinary Cd, LCd, and age in women (Table 4). For all causes of deaths, age and urinary Cd were selected as factors that significantly increased mortality risks in women. However, age and LCd were selected as factors increasing mortality from renal diseases and renal failure in women, suggesting that LCd may be a good marker to show a better dose–effect relationship with mortality risk for renal diseases, including renal failure, compared with urinary Cd.

## 4. Discussion

In a 35-year follow-up study, an increase in LCd mortality from renal diseases, particularly for renal failure, in women ≥50 years of age, living in the environmentally contaminated Kakehashi River basin in Japan, suggests a dose–response relationship between the external dose of Cd and mortality from renal diseases in women.

Previously, we conducted follow-up studies of residents of the Kakehashi River basin in Japan who had participated in a health impact survey in 1981–1982 and reported the poor life prognosis of subjects associated with an increase in urinary Cd in a 15-year follow-up study [13] and a 22-year follow-up study [14]. In particular, for renal diseases, the mortality ratio was significantly higher in women in the higher urinary Cd group, with a 10- [13] or 20-μg/g Cr cut-off level [14] compared with the lower Cd-exposure group, albeit no dose–response relationships were shown between urinary Cd and mortality risks for renal diseases. In the present study, however, a 10-μg/g Cr increase of urinary Cd increased mortality not only for all causes but also for renal diseases. 

LCd was used as an external Cd exposure maker to reflex the Cd exposure dose from the environment over a lifetime. In the present study, an increase in LCd was significantly associated with increased mortality from renal diseases in women, albeit no association was found in our previous 22-year follow-up. In women, we detected 26 deaths from renal diseases, which increased the statistical power in the present 35-year follow-up, and 14 deaths from renal diseases in the previous 22-year follow-up. Moreover, LCd was selected from two Cd exposure markers as a factor that contributed to increased mortality from renal diseases, suggesting that LCd showed a better dose–response relationship with mortality from renal diseases during a long time observation compared with urinary Cd. Since urinary Cd shows a sharp increase with increasing external Cd exposure only before the occurrence of renal tubular dysfunction [21,22], LCd might reflex the Cd dose that induces fatal renal diseases more than urinary Cd among subjects with a high prevalence of renal tubular dysfunction. 

In the Jinzu River basin, Toyama, in Japan, where itai-itai disease is endemic, an increased mortality risk associated with LCd intake was suggested in residents who participated in the health survey in 1967 [15,16]. Recently, it was reported that an increase of LCd increased the mortality for all causes, renal disease, and the toxic effects of Cd (itai-itai disease) in the 26-year follow-up of female participants to the health impact survey in the Jinzu River basin conducted in 1979–1984 [17]. These results from the Jinzu River basin are consistent with our present results, indicating that LCd dose-dependently increased mortality over long time observation, particularly for renal diseases in women, albeit their follow-up period was shorter than ours. As their mean (SD) of LCd was 3.5 (1.9) g for men and 3.0 (1.5) g for women [17] and much higher than our values, 2.7 (1.6) g for men and 2.5 (1.6) g for women, this suggests heavier Cd contamination in rice in the Jinzu River basin. Although their hazard ratio for renal diseases for a 1-g increase of LCd was 1.25 and lower than that (1.49) in our present study of the Kakehashi River basin, direct causes of deaths might be renal failure in some cases diagnosed as the toxic effects of Cd (itai-itai disease), suggesting an underestimation of the mortality risk ratio for renal diseases in the Jinzu River basin. Moreover, increased mortality associated with an increase in LCd was also observed for all causes in the Jinzu River basin in women because of the much higher rate of death for specific causes of death to Cd exposure, such as renal diseases and the toxic effects of Cd (14.1%), compared with our results in the Kakehashi River basin (3.8%).

The association between LCd and mortality for renal diseases was significant only in women in both Cd-polluted areas, suggesting that women who took Cd from rice for a long time have more risk of dying from renal failure compared with men who took Cd at the same level. Since Cd absorption is higher in women because of lower body iron stores [23,24], we compared urinary Cd, uncorrected by Cr, in each LCd category, but no significant difference was found in any categories between men and women. However, urinary-uncorrected ß2–MG increased with increasing levels of LCd and was significantly higher in women compared with men when LCd ≥ 4.5 g. These findings suggest that renal tubular dysfunction may aggravate women more than men and cause renal failure deaths in women that are highly exposed to Cd.

In the study by Nogawa et al. [17], mortality was analyzed with three covariates, including age, smoking status at present, and hypertension history, while we performed our analysis with only one covariate of age, which is a limitation of our present study. However, no effects of smoking status or hypertension history were observed on mortality from renal diseases and the toxic effects of Cd of residents of the Jinzu River basin, suggesting that the confounding factors of smoking status and hypertension history may be limited in the association between mortality from renal diseases and LCd in the women of our study. In addition, the strength of the present study may be analyzing the effects of urinary Cd and LCd and internal and external Cd exposure markers on mortality at the same time in the same population. We showed that an increase in urinary Cd also increased mortality from renal diseases in women, as well as an increase in LCd, surely indicating the fatal renal toxic effects of environmental Cd exposure on women. 

In Cd-polluted areas other than Japan, studies to investigate the dose relationship between mortality and external Cd exposure are limited. Only in Belgium, significantly higher lung cancer risk (incidence of fatal cancer) was reported for a doubling of Cd concentration in the soil of residential areas in a prospective population-based study from 1985 to 2004, which targeted the Flemish population who participated in the Cd in Belgium Study (CadmiBel) [25]. In almost all studies to investigate mortality risks for all causes [26], for cancer [27,28], and for cardiovascular diseases [29,30], internal Cd exposure markers such as urinary Cd or blood Cd were used as exposure markers because the exposure level is within background levels in their targeted population and it was difficult to estimate LCd. 

## 5. Conclusions

A dose–response relationship between the external dose of Cd over a lifetime and mortality for renal diseases was found in women living in the Cd-polluted Kakehashi River basin, suggesting clear evidence of the impact of environmental Cd exposure on increased mortality from renal diseases in women.

## Figures and Tables

**Figure 1 toxics-08-00081-f001:**
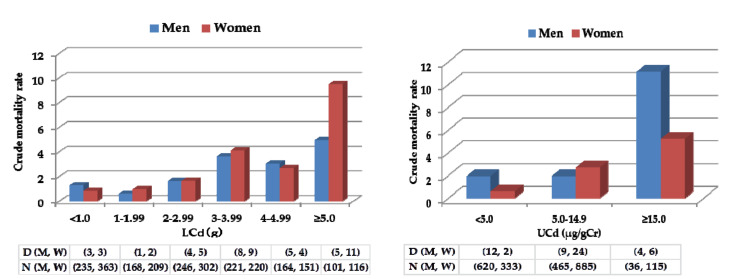
Crude mortality rates (%) for renal diseases and Cd exposure levels indicated by (**a**) LCd (+1 g) and (**b**) UCd (urinary Cd; +10 μg/g Cr). D (M, W): number of deaths for men and women; *n* (M, W): number of subjects of men and women.

**Table 1 toxics-08-00081-t001:** Levels of external cadmium (Cd) exposure and renal dysfunction examined in the 1981–1982 health impact survey and the follow-up rates and period until 2016.

Exposure and Renal Markers	Men		Women		*p*-Value
	Mean, *N*	SD, (%)	Mean, *N*	SD, (%)	
*Test results* *in 1981-2*					
*N*	1169		1433		
Age	62.5	9.1	62.9	9.4	0.281
Lifetime Cd intake (g)	2.74	1.63	2.48	1.65	0.000
Urinary Cd (μg/gCr) ^1^^,2^	4.6	1.9	7.2	1.8	0.000
Rate of Cd ≥10 (μg/gCr)	137	(11.9)	411	(29.3)	0.000
Urinary ß2–MG (μg/gCr) ^1^	166.1	6.3	252.6	6.7	0.000
Rate of ß2-MG ≥1000 (μg/gCr)	174	(14.9)	268	(18.7)	0.001
*Follow-up study*					
N (follow-up rate)	1135	(97.1)	1361	(95.0)	
Follow-up period (months)	222	123.4	256	128.4	0.000

^1^ Geometrical mean and geometrical standard deviation, ^2^ Participants were 1121 men and 1333 women. N: number of subjects, SD: standard deviation.

**Table 2 toxics-08-00081-t002:** Dose–response relationships between hazard ratios and Cd exposure markers in men.

Causes of Deaths	D	HR	95%CI	*p*-Value
*Urinary Cd (+10 μg/gCr)*				
All causes	890	1.35	1.06, 1.71	0.014
Malignant neoplasms	253	1.17	0.74, 1.84	0.496
Cardiovascular diseases	125	1.39	0.74, 2.61	0.300
Cerebrovascular diseases	117	1.17	0.61, 2.23	0.633
Respiratory diseases	151	1.07	0.60, 1.90	0.819
Renal diseases	25	2.94	0.70, 12.3	0.140
Renal failure	22	2.73	0.59, 12.7	0.202
*Lifetime Cd Intake (+1 g)*				
All causes	903	0.97	0.94, 1.02	0.210
Malignant neoplasms	254	0.98	0.90, 1.06	0.417
Cardiovascular diseases	129	0.98	0.88, 1.09	0.680
Cerebrovascular diseases	118	0.92	0.83, 1.03	0.148
Respiratory diseases	152	0.94	0.85, 1.04	0.214
Renal diseases	26	1.20	0.94, 1.52	0.141
Renal failure	23	1.12	0.87, 1.43	0.390

D: number of deaths, HR: hazard ratio, CI: confidence interval, N: number of subjects *n* = 1121 for urinary Cd, *n* = 1135 for lifetime Cd intake.

**Table 3 toxics-08-00081-t003:** Dose–response relationships between hazard ratios and Cd exposure markers in women.

Causes of Deaths	D	HR	95% CI	*p*-Value
*Urinary Cd (+10 μg/gCr)*				
All causes	873	1.34	1.04, 1.71	0.019
Malignant neoplasms	186	1.18	0.66, 2.10	0.572
Cardiovascular diseases	157	1.31	0.76, 2.26	0.330
Cerebrovascular diseases	132	1.05	0.55, 1.98	0.894
Respiratory diseases	109	1.25	0.66, 2.35	0.490
Renal diseases	32	5.23	1.97, 13.9	0.001
Renal failure	23	5.12	1.65, 15.9	0.005
*Lifetime Cd Intake (+1 g)*				
All causes	889	0.99	0.95, 1.03	0.696
Malignant neoplasms	187	0.97	0.88, 1.06	0.460
Cardiovascular diseases	160	1.00	0.92, 1.10	0.970
Cerebrovascular diseases	133	1.00	0.90, 1.11	0.971
Respiratory diseases	113	0.90	0.80, 1.01	0.067
Renal diseases	34	1.49	1.20, 1.85	0.000
Renal failure	23	1.48	1.14, 1.93	0.003

D: number of deaths, HR: hazard ratio, CI: confidence interval, N: number of subjects. *n* = 1333 for urinary Cd, *n* = 1361 for lifetime Cd intake.

**Table 4 toxics-08-00081-t004:** Dose–response relationships between hazard ratios and Cd exposure markers in women.

Causes of Deaths	HR	95% CI	*p*-Value
All causes			
Age	1.15	1.14, 1.16	0.000
Urinary Cd (+10 μg/gCr)	1.33	1.04, 1.70	0.021
Renal diseases			
Age	1.12	1.07, 1.18	0.000
Lifetime Cd Intake (+1 g)	1.46	1.17, 1.82	0.001
Renal failure			
Age	1.18	1.10, 1.26	0.000
Lifetime Cd Intake (+1 g)	1.49	1.14, 1.93	0.003

D: number of deaths, HR: hazard ratio, CI: confidence interval, Number of subjects = 1333.

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
