# Peer review of "Lifetime Cadmium Exposure and Mortality for Renal Diseases in Residents of the Cadmium-Polluted Kakehashi River Basin in Japan"

_toxics, 2020, doi:10.3390/toxics8040081_

Round 1

Reviewer 1 Report

This is an interesting paper dealing with human exposure to Cd. However, a couple of issues, namely in the absence of control subjects and lack of proper confounder analysis ,raise questions concerning the relevance and soundness of the analysis.

I have identified some issues below that I think the authors should address before the manuscript can be considered for publishing.

How was the follow-up conducted? Was it a prospective follow up with periodic gathering of data from the population? Was it a single follow-up after 35-years? It is not clear so please explain with more detail.

How was the Hazard Ratio calculated? The authors need to provide these details in the methods.

The authors need to make comparisons with low Cd exposure group. How did kidney failure prevalence vary in these subjects? How does this low Cd exposure population in both sexes compare with the high Cd population? This is particularly important since normally men show a higher prevalence of CKD than women and it assumes particular relevance in this study since the only confounder the authors controlled for was age and kidney disease may results from other health conditions such as smoking and hypertension…

Women having higher Cd intake is than men is expected and is largely related to the particularities of Cd toxicokinetics, namely the differences between sexes in gut absorption. The authors should explain this in the manuscript to allow comprehension by a reader less familiar with Cd toxicology.

How do you explain that the association between urinary Cd>10 and all causes of death is equal in both men and women, but only in women it is possible to single out a cause of death (Kidney failure) ?

Assuming the same lifetime intake of rice for both men in women may introduce a significant bias in the results. The caloric nedds and hence food intake is generally higher for men than women. What was the rationale behind using the same rice intake value for both sexes (333.5 g per day)? Where did the authors base their choice of intake value?

Table 1 Lifetime Cd (g) for men is 2.7 ± 1.6 and for women 2.5 ± 1.6. This is largely equivalent, but still the statistical differences are very significant (p<0.0001) which is strange. The authors should provide as supplementary material the individual values for LCd so that readers can judge appropriately the differences between sexes.

Author Response

Comment 1: How was the follow-up conducted? Was it a prospective follow up with periodic gathering of data from the population? Was it a single follow-up after 35-years? It is not clear so please explain with more detail.

Reply: We conducted a prospective follow-up survey and determined their survival status (alive or dead) by collection family registry records from city office. Therefore, line 91 and 92-93, page 3, we added some words to explain method more clearly. Regarding methods to determine causes of deaths, more explanation was added on line 94-95, page 3.

In facts, we performed several follow-up surveys for 8 years, 15 years, and 22 years before the present survey. However, this is the first time to determined causes of deaths by data linkage with national vital statistics data (previously, we used death certificates in Public Health Office). Therefore, we didn’t mention about previous follow-up surveys in method section.

Comment 2: How was the Hazard Ratio calculated? The authors need to provide these details in the methods.

Reply: We added explanation for Cox’s proportional hazard analysis on line103-5, page 3.

Comment 3: The authors need to make comparisons with low Cd exposure group. How did kidney failure prevalence vary in these subjects? How does this low Cd exposure population in both sexes compare with the high Cd population? This is particularly important since normally men show a higher prevalence of CKD than women and it assumes particular relevance in this study since the only confounder the authors controlled for was age and kidney disease may result from other health conditions such as smoking and hypertension…

Reply: We added 1 result section (3.3 Dose response relationships……) with Figure 1 illustrating dose-response relationships between Cd exposure levels and crude mortality rate for renal diseases, showing mortality rates, nearly equal to renal disease prevalence, including those in low and high exposure groups in both sexes (line 124-138, page 4-5 . As you suggested, in general population, prevalence of CKD is higher in men than that in women. In the present subjects, mortality for renal diseases was higher in men in the lowest exposure group indicating by LCd or urinary Cd, suggesting other factors than Cd exposure. While, we can find more deaths for renal diseases in the highest LCd group (LCd >= 5g) in women, compared in men. We described these in the text on line 131-133, page 5.

Comment 4: Women having higher Cd intake is than men is expected and is largely related to the particularities of Cd toxicokinetics, namely the differences between sexes in gut absorption. The authors should explain this in the manuscript to allow comprehension by a reader less familiar with Cd toxicology.

Reply: As you suggested, women can absorb Cd through gut, because of less iron stored compared with men. Therefore, internal Cd exposure marker such as urinary Cd, might be higher than that in men with the same level of LCd. However, our comparisons of urinary Cd (uncorrected by creatinine) between men and women in each LCd category showed significant difference of urinary b2-MG in higher LCd levels, but no difference of urinary Cd. These findings suggest that aggravation of renal tubular dysfunction can be happen and cause renal failure deaths in women more than men. We described them in an additional paragraph in discussion section, line 191-9, page 6.

Comment 5: How do you explain that the association between urinary Cd>10 and all causes of death is equal in both men and women, but only in women it is possible to single out a cause of death (Kidney failure) ?

Reply: In men, relationships between urinary Cd and mortality for renal diseases and renal failure were not significant, but mortality risk ratios were more than 2, suggesting higher risks. As we indicated a graph to indicate dose-response relationship, increased mortality was observed only in men with urinary Cd ≥ 15, while mortality increased in proportion to urinary Cd in women. Also, less significant result may be partly caused by the small number of men with urinary Cd ≥ 15 (only 36 men, less statistical power) . Taken together, relationships between urinary Cd and mortality are similar for all causes and renal diseases in men.

Comment 6: Assuming the same lifetime intake of rice for both men in women may introduce a significant bias in the results. The caloric nedds and hence food intake is generally higher for men than women. What was the rationale behind using the same rice intake value for both sexes (333.5 g per day)? Where did the authors base their choice of intake value?

Reply: We used to estimate individual LCd using equation suggested by Nogawa et al. (1989). They used 333.5 g per day for rice ingestion amount for a resident in Kakehashi River basin reported in the food survey conducted by prefecture government. That value of daily rice ingestion was obtained by duplicated diet method to collect all meals per day for selected households in each community in 1975-6 (reported in 1978). As you suggested, actual rice ingestion is different between sexes, but this is the best value for rice ingestion, because data was examined in the targeted area and time when Cd problem had just discovered. To add more explanation of examination method, we added a phrase “obtained by duplicated diet method” in method section on line 73, page 2.

Comment 7: Table 1 Lifetime Cd (g) for men is 2.7 ± 1.6 and for women 2.5 ± 1.6. This is largely equivalent, but still the statistical differences are very significant (p<0.0001) which is strange. The authors should provide as supplementary material the individual values for LCd so that readers can judge appropriately the differences between sexes.

Reply: We showed the number of subjects in each LCd category in men and women in Figure 1, showing more number of women belong to lower categories compared with men. Also, mean and SD were calculated in mg and finally shown in g in the table. So, we showed more decimal of mean and SD values of LCd, indicating clear difference.

Reviewer 2 Report

This research focuses on the exposure and health risks of cadmium pollution from the environment to residents' dietary intake.

In addition to cadmium pollution and health risks, many studies have revealed that exposure to arsenic (As) compounds may cause kidney damage. Intake of inorganic arsenic may increase the risk of skin cancer, liver cancer, bladder cancer and lung cancer. IARC also confirmed that inorganic arsenic is carcinogenic to humans. Blood, urine, hair and nails can all be used to measure arsenic levels. Urine test is the most reliable test item.

Since the arsenic in rice has also been concerned from some countries and WHO/FAO in recent years, please indicate whether there are monitoring data of arsenic content in rice and in urine. If the content of arsenic in the environment and rice products of the Kakehashi River basin is low and stable, we may conclude that the influence of arsenic on this study can be ignored.

Please describe the method used in this study to analyze the content of cadmium in urine. I believe the current method should be different from the earlier method for Cd analysis.

Suggest to amend the title as “lifetime Cd exposure and mortality for renal diseases in residents of the cadmium polluted Kakehashi river basin in Japan”.

Author Response

Comment 1: In addition to cadmium pollution and health risks, many studies have revealed that exposure to arsenic (As) compounds may cause kidney damage. Intake of inorganic arsenic may increase the risk of skin cancer, liver cancer, bladder cancer and lung cancer. IARC also confirmed that inorganic arsenic is carcinogenic to humans. Blood, urine, hair and nails can all be used to measure arsenic levels. Urine test is the most reliable test item.

Reply: The present study area, Kakehashi River basin in Komatsu city, Japan, is irrigated by Kakehashi River, upper reach of which a Cu mine was operated from 1930 until 1971. The health impact survey in residents of Kakehashi River basin was started in 1970’s, but pollution of Kakehashi River had been known after mining activity started , because of no fish in the river. Therefore, the survey of heavy metals in sediments of canals from the river, soil of paddy fields, and rice from their fields were examined to evaluate environmental pollution in this area. On the other hand, in Japan, we have an environmental standard for human health and prefecture government needs to examine As in drinking water and river water including Kakehashi River since 1971. In this area, local government examined As in well water and river, but no positive results was obtained (less than standard). Moreover, we found high level of heavy metals, particularly Cd in soil and rice, and found that renal dysfunction indicated by protein, glucose, and b2-MG in urine associated with Cd in rice and LCd. Therefore, in this area, As seems to be not a cause of renal dysfunction.

Comment 2: Since the arsenic in rice has also been concerned from some countries and WHO/FAO in recent years, please indicate whether there are monitoring data of arsenic content in rice and in urine. If the content of arsenic in the environment and rice products of the Kakehashi River basin is low and stable, we may conclude that the influence of arsenic on this study can be ignored.

Reply: In general, Cd in rice is high and As is stable in Japan except some area in south island. Most important souse to increase As intake is sea weed in Japan. Therefore, we have no data As in rice in these days in 1970s. Recently, in some areas in Japan, they tried to decrease Cd in rice by controlling irrigation water, but found that their treatment increased AS in rice. However, this trial was not done in our area and we can ignore effects of As on human health in this area.

Comment 3: Please describe the method used in this study to analyze the content of cadmium in urine. I believe the current method should be different from the earlier method for Cd analysis.

Reply: We added explanation of urinary Cd measurement in method section with a reference, line 85-88, page 3.

Comment 4: Suggest to amend the title as “lifetime Cd exposure and mortality for renal diseases in residents of the cadmium polluted Kakehashi river basin in Japan”.

Reply: As you suggested, we changed the title.

Reviewer 3 Report

Comments and suggestions for the authors

Nishijo and colleagues investigated the dose-response relationships between Cadmium (Cd) exposure and mortality, focusing on renal diseases. They included participants of the 1981-1982 health impact survey and assessed the outcomes of interest after a 35-year follow-up. Additionally, they calculated lifetime Cd intake (LCd) using a formula Cd in rice levels.

The introduction clearly explains that the markers used are good exposure markers for cadmium exposures and the authors already published on this topic. The introduction could be improved by adding a paragraph on the known effects of on human health.

The authors briefly explain the formula to calculate LCd however I believe that more in depth explanation is needed to convince the reader that this is a solid method. For instance, the formula takes into consideration the time that a participant lives in a non-exposed area but not what the weight is or which part of their life they lived in that area. I assume that participants that lived in the Cd contaminated area during an important part of their life, for instance early childhood, puberty or during pregnancy.

The authors conclude in the first part of their discussion that increase of LCD in associated with mortality of renal disease in residents of the contaminated area, however from their results it seems that this was only the case in women; no association was seen in men. Please revise this statement.

For the discussion, a paragraph on the limitations of the study is missing.  

In the 5th paragraph of the discussion, (line 174- 183), the authors refer to a previously published study. Please clarify which study is meant.

Author Response

Comment 1: The introduction clearly explains that the markers used are good exposure markers for cadmium exposures and the authors already published on this topic. The introduction could be improved by adding a paragraph on the known effects of on human health.

Reply: We added a paragraph to briefly describe Cd effects on human health at the top of introduction section, line 33-36, page 1.

Comment 2: The authors briefly explain the formula to calculate LCd however I believe that more in depth explanation is needed to convince the reader that this is a solid method. For instance, the formula takes into consideration the time that a participant lives in a non-exposed area but not what the weight is or which part of their life they lived in that area. I assume that participants that lived in the Cd contaminated area during an important part of their life, for instance early childhood, puberty or during pregnancy.

Reply: Calculation of LCd in the present study is not perfect and improvement to take a count for difference along life time is necessary. However, many previous studies showed that it is a good marker for dose-response relationship with renal tubular dysfunction with population more than 1000 subjects. Also, we used urinary Cd, which is a good exposure marker. Therefore, for this population, we believe that this calculation is best way.

Comment 3: The authors conclude in the first part of their discussion that increase of LCD in associated with mortality of renal disease in residents of the contaminated area, however from their results it seems that this was only the case in women; no association was seen in men. Please revise this statement.

Reply: We revised this sentence; “residents” to “women” and added “ in women” at the end of this sentence, line 151 and 153, page 5.

Comment 4: For the discussion, a paragraph on the limitations of the study is missing.

Reply: In the 5th paragraph, we described our limitation; our covariate is only age.

Comment 5: In the 5th paragraph of the discussion, (line 174- 183), the authors refer to a previously published study. Please clarify which study is meant.

Reply: We changed expression from “In their study” to “In the study investigated Nogawa et al. [17]” to make it clear which reference we used, line 200, page 6.

Round 2

Reviewer 1 Report

The authors have addressed the issues raised but I still have some issue

Figure 1 is supposed to be a dose-reponse analysis but I see no regression or correlation data. Please include in figure and text.

The paragraph from lines 131-138 explaining variations in Fig1 between men and women  is a bit confusing. Actually the reply to my comment about this is better. Please explain that in men CKD death rates are normally higher than in women and this can be attributed to several risk factors which you did not control for.

The authors included a sentence (lines 85-88) concerning Cd analysis in urine but they do not mention analysis of any certified reference material or any other analysis quality control parameter. This has to be provided so that the reader can judge the soundness of instrumental analysis.

Written english needs a review by a native speaker

Author Response

Comment 1: Figure 1 is supposed to be a dose-response analysis but I see no regression or correlation data. Please include in figure and text.

Reply: Figure 1 shows only crude mortality rates in each exposure groups according to LCd and urinary Cd in men and women (no analytical results of dose-response relationship). Therefore, we changed subtitle 3.3 (line 126) and the first sentence for explanation of Fig. 1 (line 127-8, page 4).

Comment 2: The paragraph from lines 131-138 explaining variations in Fig1 between men and women is a bit confusing. Actually, the reply to my comment about this is better. Please explain that in men CKD death rates are normally higher than in women and this can be attributed to several risk factors which you did not control for.

Reply: As you suggested, we replaced these sentences to those in our previous reply to your comment with additional explanation (line 132-9, page 5).

Comment 3: The authors included a sentence (lines 85-88) concerning Cd analysis in urine but they do not mention analysis of any certified reference material or any other analysis quality control parameter. This has to be provided so that the reader can judge the soundness of instrumental analysis.

Reply: As you suggested, we added the sentence that we used freeze-dried standard reference material for testing accuracy and precision of the analytical method (line 87-9, page 3).

Thank you very much for your meaningful comments.